# Molecular Characterization of HOXA2 and HOXA3 Binding Properties

**DOI:** 10.3390/jdb9040055

**Published:** 2021-12-03

**Authors:** Joshua Mallen, Manisha Kalsan, Peyman Zarrineh, Laure Bridoux, Shandar Ahmad, Nicoletta Bobola

**Affiliations:** 1School of Medical Sciences, University of Manchester, Manchester M13 9PT, UK; joshua.mallen@manchester.ac.uk (J.M.); peyman.zarrineh@manchester.ac.uk (P.Z.); 2School of Computational and Integrative Sciences, Jawaharlal Nehru University, New Delhi 110067, India; manisha8013@gmail.com (M.K.); shandar@jnu.ac.in (S.A.); 3Louvain Institute of Biomolecular Science and Technology, Université Catholique de Louvain, 5 (L7.07.10) Place Croix du Sud, 1348 Louvain-la-Neuve, Belgium; bridoux.laure@gmail.com

**Keywords:** transcription factor, HOX, development, branchial arches

## Abstract

The highly conserved HOX homeodomain (HD) transcription factors (TFs) establish the identity of different body parts along the antero–posterior axis of bilaterian animals. Segment diversification and the morphogenesis of different structures is achieved by generating precise patterns of HOX expression along the antero–posterior axis and by the ability of different HOX TFs to instruct unique and specific transcriptional programs. However, HOX binding properties in vitro, characterised by the recognition of similar AT-rich binding sequences, do not account for the ability of different HOX to instruct segment-specific transcriptional programs. To address this problem, we previously compared HOXA2 and HOXA3 binding in vivo. Here, we explore if sequence motif enrichments observed in vivo are explained by binding affinities in vitro. Unexpectedly, we found that the highest enriched motif in HOXA2 peaks was not recognised by HOXA2 in vitro, highlighting the importance of investigating HOX binding in its physiological context. We also report the ability of HOXA2 and HOXA3 to heterodimerise, which may have functional consequences for the HOX patterning function in vivo.

## 1. Introduction

Changes in gene expression drive embryonic development. Transcription factors (TFs) control gene expression by binding to regions in the genome [1,2]. TFs contain DNA binding domains which recognize DNA in a sequence-specific manner. However, as most TF-binding sites are short (usually 6–12 bases), these interactions are typically insufficient to direct a TF to its functional targets. As a result, predicting gene regulation from TF-binding motifs remains a major practical challenge. This is exemplified by homeodomain (HD) TFs, composed of hundreds of TFs sharing a conserved DNA binding domain [3,4]. HD display highly similar sequence recognition properties and bind the same core of the four-base-pair sequence TAAT [5].

The highly conserved HOX HD TFs provide an ideal model to explain how TFs select their target enhancers to direct specific transcriptional programs in vivo. HOX TFs establish the identity of different body parts along the antero–posterior (AP) axis of bilaterian animals [6,7]. In mammals, there are 39 *Hox* genes, grouped into anterior (HOX1-2), central (HOX3–8), and posterior (HOX 9–13) paralog groups [8]. HOX paralogs occupy sequential positions along the chromosome [9], which translates into sequential HOX expression at different levels of the AP axis, conferring specific spatial (and temporal) coordinates to each cell. Sharp patterns of HOX differential expression along the AP axis, combined with the ability of different HOX to control the expression of unique and specific sets of genes, results in segment diversification and the morphogenesis of different structures along the AP axis. However, we still do not understand how HOX achieve the required specificity to instruct segment-specific transcriptional programs by recognising similar binding sequences in vitro. To investigate this, we have focussed on Hoxa2 and Hoxa3, which instruct the morphogenesis of the branchial arches (BA), a series of transient, repetitive blocks of embryonic tissues that will merge to form the face and the neck in vertebrates. HOXA2 is expressed in the second branchial arch (BA2), which is also the domain most affected by *Hoxa2* inactivation in mice [10,11]. In the BA2, HOXA2 regulates a network of TFs, including the HD proteins MEIS2 [12], MEOX1 [13], HMX1 [14], SIX2 [15] and the zinc finger transcriptional repressors ZFP703 and ZFP503 [12]. The following BA along the AP axis, the BA3, is patterned by HOX paralogs 3 [16]. HOXA2 remains expressed in BAs posterior to BA2, raising the question of how posterior programs prevail over more anterior ones when different paralogs (HOXA2 and HOXA3) are co-expressed in the same cells. We previously identified three main determinants of HOX paralog 2 and 3 binding: the ability of HOXA3 to recognize unique variants of the HOX core motif, a HOXA2 higher affinity for shared HOX motifs, and the cooperation with TFs that are differentially expressed across the BAs [17]. Here, we ask if the enrichment of specific variants of the core HOX motif, observed in HOX binding in vivo, can be solely explained by different HOX binding affinities. We find that the top motif enriched in high confidence HOXA2 peaks is not recognised by HOXA2 in vitro. We also report that HOXA2 and HOXA3 can be found in the same protein complex; this provides a possible molecular explanation for how posterior programs prevail over more anterior ones when different paralogs (HOXA2 and HOXA3) are co-expressed in the same cells.

## 2. Material and Methods

### 2.1. Co-Immunoprecipitation

Human Embryonic Kidney (HEK)293T cells were seeded in 6-well plates at 400,000 cells/well and transfected after 24 h with FuGENE^®^ 6 (E2691, Promega, Madison, WI, USA) (666 ng HA, 666 ng FLAG, and either 333 ng each of MEIS and PBX or 666 ng of empty pcDNA3 vector per condition). Proteins were collected at 48 h post-transfection. CoIP was performed using Dynabeads™ Protein G (10004D, Thermo Fisher Scientific, Waltham, MA, USA) and Monoclonal ANTI-FLAG^®^ M2 antibody (F1804, Sigma, UK).

### 2.2. Electrophoretic Mobility Shift Assays (EMSA)

The Meis2 fluorescent probes were made by PCR using ATTO700-labelled primers (Eurofins, Germany). TNT proteins were made using TnT^®^ Quick Coupled Transcription/Translation System (L1170, Promega, Madison, WI, USA). 2 μL TNT protein were mixed with 16 ng purified probe, 2 μg poly dIdC, 4% Ficoll, 20 mM HEPES, 39 mM KCl, 1 mM DTT, and 0.1 mM EDTA, flicked gently to mix, incubated for 12 min, and run on a 4% non-denaturing acrylamide gel (pre-run for 30 min) for 90 min at 70 V and in dark conditions. Where indicated, 1 μL HOXA2/HOXA3 was incubated for 5 min prior to the addition of 1 μL PBX/MEIS and the probe, followed by a 12-min incubation. Gels were imaged using Odyssey CLx (LI-COR, Lincoln, NE, USA) and quantified using ImageJ 1.52a.

### 2.3. Sequence Analysis

To explore the preferential differences that can be attributed to TGATGGAT, HOXA2 summits were divided into two sets on the basis of the presence of the TGATGGAT motif (motif positive or motif+) or no motif (motif negative or motif−). Regions were extended for ±200 bp from the centre of the TGATNNAT, and the genomic sequences were extracted using bedtools [18]. After computing the GC contents of every Hoxa2 bound sequence, we sorted them in the order of decreasing GC contents (within ±10 bp of the centre) and observed the motifs generated in the top and last 100 GC-sorted sequences using *MEME* [19]. For each of the sequences generated as described above, the motif occurrences in the flanking regions were calculated using known motifs in the Homer database [20].

## 3. Results and Discussion

### 3.1. Changes to the Core Variable Sequence Affect HOXA2 Binding In Vitro

Both HOXA2 and HOXA3 bind the sequence TGATNNAT [17], which is recognised in a complex with three amino acid loop extension (TALE) HD TFs PBX and MEIS [4,21,22]. In vivo, HOXA2 and HOXA3 extensively binds with TALE TFs [12,17,23]. The TGATNNAT or HOX-PBX motif contains a variable core; the counting instances of TGAT**NN**AT in top HOX peaks shows that HOXA2 peaks are enriched in TGAT**GG**AT, followed by TGAT**TG**AT and TGAT**TA**AT or TGAT**TT**AT (Figure 1A).

To assess if this differential enrichment reflects a diverse binding affinity of HOXA2 for the core variable regions, we tested HOXA2 binding to mutant versions of the *Meis2* enhancer [17]. The *Meis2* probe, which spans the summit of a high confidence HOXA2 peak, contains only one HOX-PBX site (TGATTGAT). The binding of HOXA2 in a complex with PBX and MEIS to the *Meis2* probe is abolished by mutagenesis of the HOX-PBX motif (where TGATTGAT is changed to TGCGTGCG) [17]. We designed and tested four variants of the *Meis2* probe; these mutant variants are distinguished only by changes in the core variable nucleotides of the TGATNNAT site, while the remainder of the sequence is unaltered. Specifically, we replaced the TG (wild-type) with GG (top enriched), TA (3rd enriched) and CG (no enrichment) (Figure 1B). For all the mutant probes tested, we did not observe the formation of additional nonspecific bands relative to the wild-type probe (Appendix A). We found that the HOXA2 ability to bind the *Meis2* probe with PBX/MEIS was strongly decreased when TG was changed to GG or CG (7–9 times lower) (Figure 1C). As expected from motif enrichment (Figure 1A), introducing a C in the core motif (TG → CG) greatly reduced the formation of the HOXA2/PBX/MEIS complex (trimeric complex) relative to the wild-type probe (Figure 1C,D); this was also accompanied by a strong reduction in the formation of PBX/MEIS dimer (Appendix A). Similarly, the formation of the HOXA2/PBX/MEIS complex (trimeric complex) was strongly reduced by introducing a G (TG → GG). This reduction appears to be mainly caused by the inability of HOXA2 to bind with PBX and MEIS, as PBX/MEIS binding as a dimer was only mildly affected (Figure 1C,D and Appendix A). Additional sequences to the variable core may be required for HOXA2 binding to GG containing motifs, or, alternatively, HOXA2 may bind this site in vivo with other partners than PBX/MEIS. Our knowledge on HOX binding derives largely from experiments conducted in vitro [21]. HOX recognition sequences in vitro are not necessarily present on HOX functional enhancers, which display low affinity ‘versions’ of the sequences recognised in vitro [24]. Our results show a large discrepancy between HOXA2 binding in vivo and in vitro, highlighting the importance of understanding HOX molecular function in its physiological context.

Perhaps the most striking effect was observed when introducing a change from TG → TA. We observed a strong monomeric band, indicating that HOXA2 on its own displays a high affinity to this version of the *Meis2* probe (Figure 1C and Figure 2D). Adding PBX/MEIS generated a comparable trimeric band relative to the *Meis2* wild type but did not eliminate HOXA2 binding as a monomer. Relative to the wild type and other variants, the *Meis2* probe containing the TGATTAAT sequence displays the full binding configuration, monomer (A2), dimer (PBX/MEIS), and trimer (A2/PBX/MEIS) (Figure 1C).

These observations indicate that nucleotide changes in the core variable sequence have a profoundly different effect on HOXA2 binding to the *Meis2* probe, either by affecting PBX/MEIS binding (CG), HOXA2 binding with PBX/MEIS (GG), or HOXA2 binding as a monomer (TA). They also suggest that the assembly of a PBX/MEIS complex on DNA is a requirement for the formation of the HOXA2 trimeric complex: in all cases, we observed a displacement of the dimeric complex when adding HOXA2. Limiting amounts of PBX/MEIS in the reaction are an unlikely explanation for this effect, since we observed a similar displacement of the dimer even in conditions in which less PBX/MEIS were bound to DNA (TA probe) (more PBX and MEIS freely available). In addition to increasing the specificity, cooperative HD TF binding can alter regulatory outcomes [25]. It will be interesting to test if HOXA2 binding on its own or in a complex with PBX/MEIS could mediate opposing outcomes in gene expression (e.g., repression versus activation).

As suggested by the variants enrichment in HOXA3 top peaks [17], HOXA3 appears to be more promiscuous (Figure 2A). Changing the variable core to GG or CG decreased the formation of the trimeric complex, but to a lesser extent than what was observed with A2 (Figure 2B,C). Introducing TA in the variable core resulted in HOXA3 binding as a monomer (Figure 2B,D). Moreover, all mutations appeared to favour the formation of a HOXA3 trimeric complex relative to HOXA2 (Appendix A).

### 3.2. Characterisation of HOXA2-Bound TGATGGAT Motif

The motif TGATGGAT is the top enriched variant in high confidence HOXA2 peaks, but it is bound with a low affinity by HOXA2 in a complex with PBX and MEIS. To explain this discrepancy and identify additional sequence properties that may facilitate HOXA2 binding, we explored sequences flanking the TGATGGAT motif in HOXA2 peaks. First, we explored the rigidity of HOXA2 peaks containing TGATGGAT (motif+) versus peaks that did not contain TGATGGAT (motif−). Rigidity is measured in terms of GC contents, where a high GC content denotes more stable or rigid regions, while a low GC content indicates more flexible regions. We centred HOXA2 peaks on the TGATNNAT motif, computed the GC contents of every HOXA2 peak, and sorted peak summits in the order of decreasing GC contents (within ±10 bp of the centre). A de novo motif search of the top 100 (high GC) peaks identified TGATGGAT as the most enriched motif, while the last 100 peaks (low GC) were enriched in the TGATTGAT motif (Figure 3A,B).

Next, we looked for the occurrence of TGATGGAT in peaks with the highest and lowest GC content (100 sequences on both extremes). Indeed, we found that HOXA2 summits containing TGATGGAT primarily (92% of all motif+) came from GC-rich regions, whereas TGATGGAT-negative HOXA2 summits tended to be dominated by GC-depleted sequences (64% of all motif− sites) (Figure 3C). Our results suggest that TGATGGAT-containing HOXA2 peaks form more stable structures compared to the remaining HOXA2 peaks.

Next, we asked if the binding of HOXA2 to TGATGGAT-containing regions was mediated by cooperation with TFs other than MEIS and PBX. To address this question, we assessed the enrichment of TF binding sites in HOXA2 motif+ peaks, by comparing Hoxa2 motif+ and Hoxa2 motif− for TF binding sites, using different distance ranges. A distance range of 50 bp from the centre was found to be most informative. We found 16 TF recognition motifs that were exclusive to HOXA2 motif+ sequences (i.e., none of these motifs was enriched in any of the 1439 Hoxa2 motif− sequences). Specifically, we found that 264 HOXA2 peaks out of the 386 motif+ peaks contained at least one of these 16 TFs. The frequencies of occurrence for the individual TFs in this list are provided in Figure 3D. The most highly enriched motif around the TGATGGAT corresponds to SMAD3 recognition sites, suggesting that HOXA2 may cooperate with mediators of BMP signalling at these sites.

### 3.3. HOXA2 and HOXA3 Are Found in the Same Complex with TALE

HOX TFs display a general posterior prevalence rule, whereby when multiple HOX are expressed in a segment, the most posterior HOX has the strongest effect on segment development [9]. HOXA2 is highly expressed in the BA2 and at lower levels in the BA3, which is under the control of HOX paralogs 3 (including HOXA3) [17]. We previously found that increasing levels of HOXA3 could compete with and dampen HOXA2 transcriptional programmes [17]. Here, we investigate another possible mechanism of posterior prevalence, namely the possibility that HOXA3 could impede HOXA2 access to its binding sites in the genome. We found that both HOXA2 and HOXA3 interacted with MEIS1 and MEIS2 in co-precipitation experiments [17]. Starting from this observation, we next asked if HOXA2 and HOXA3 could be found in the same protein complex. By overexpressing HOXA2 and HOXA3 in HEK cells, followed by CoIP, we found HOXA2 and HOXA3 in a protein complex (Figure 4A).

The interaction was highly increased by the addition of MEIS and PBX, indicating that HOXA2 and HOXA3 could interact with each other in the presence of MEIS and PBX. However, when adding co-translated HOXA2 and HOXA3 with PBX and MEIS to the *Meis2* probe, we did not observe the formation of any super-shift, arguing against the ability of the tetrameric complex to bind DNA [17]. We reasoned that, given the higher affinity of HOXA2 for the *Meis2* enhancer relative to HOXA3, a higher HOXA3 concentration may be required for the formation of a favourable quaternary complex on DNA. This would also mimic the physiological condition, in the BA3, where HOXA3 is expressed at a higher level than HOXA2 [17]. Therefore, we repeated the experiments by adding a higher concentration of HOXA3. We found that adding HOXA3 at a higher concentration resulted in the formation a HOXA3 trimeric band, but no higher band corresponding to the tetrameric complex appeared (Figure 4B, quantifications in Appendix A). These observations suggest that, if formed in vitro, this complex is unable to bind DNA. The same results were observed on the *Meis2* variants tested in Figure 1 (Figure 4C, quantifications in Appendix A), indicating that there is no detectable tetrameric binding to DNA on a range of sequences tested.

The interaction of HOXA3 with HOXA2 provides a possible explanation for the posterior prevalence rule: trapping HOXA2 in a nonfunctional or non-DNA binding complex may enable HOXA3 to outcompete HOXA2 and drive a HOX3-specific programme in BA3. Additional examples of HOX heterodimerisation have been reported in the literature, suggesting that this could be a general mechanism used by the more posterior HOX to neutralise the activity of more anterior ones [26].

## 4. Conclusions

Our knowledge on HOX binding derives largely from experiments conducted in vitro. We have assessed if the HOXA2 binding ability in vivo mirrors HOXA2 binding in vitro. Unexpectedly, we find that the most frequent motif in high confidence HOXA2 peaks is bound by HOXA2 (in complex with PBX/MEIS) with a low affinity. The observed discrepancy in HOXA2 binding in vivo and in vitro highlights the importance of investigating HOX binding in its physiological context.

We also uncover the ability of HOXA2 and HOXA3 to heterodimerise in a complex with PBX/MEIS; trapping HOXA2 in a complex may limit HOXA2 binding to its targets in domains controlled by posterior HOX, ultimately promoting the HOX paralog 3 transcriptional programme in BA3.

## Figures and Tables

**Figure 1 jdb-09-00055-f001:**
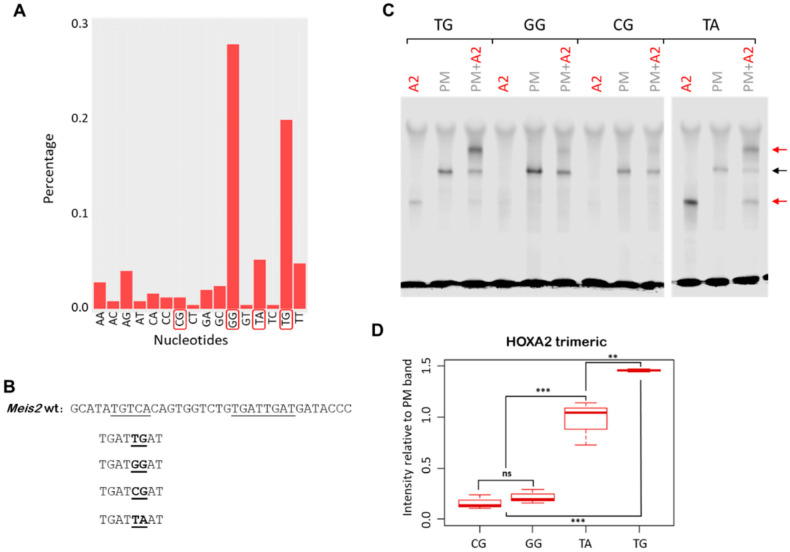
HOX binding affinity for in vivo bound HOX-PBX sites. (**A**) Percentage of TGATNNAT variants (all possible permutations of the NN variable region) in top 250 HOXA2 peaks, ranked by fold enrichment. (**B**) Reverse complement section of the *Meis2* probe sequence containing HOX-PBX and MEIS motifs, underlined, and the mutant sequences with the relevant substitutions in bold (wild-type sequence is TGATTGAT). (**C**) HOXA2 binding to the *Meis2* probes containing mutations detailed in (**B**), in the presence and absence of PBX/MEIS (PM). Trimeric complex binding is reduced in all mutants, but TA mutant sees an increase in the binding of monomeric HOXA2. Black arrows indicate PBX/MEIS dimer, red arrows indicate HOXA2 interactions with the probe. (**D**) The quantification of HOXA2/PBX/MEIS bands from experiment in (**C**) (three replicates) shows a reduction in all mutants relative to the wild type. All bands are quantified relative to the wild-type PBX/MEIS band. ** *p*-value < 0.005; *** *p*-value < 0.0005; ns: no significant.

**Figure 2 jdb-09-00055-f002:**
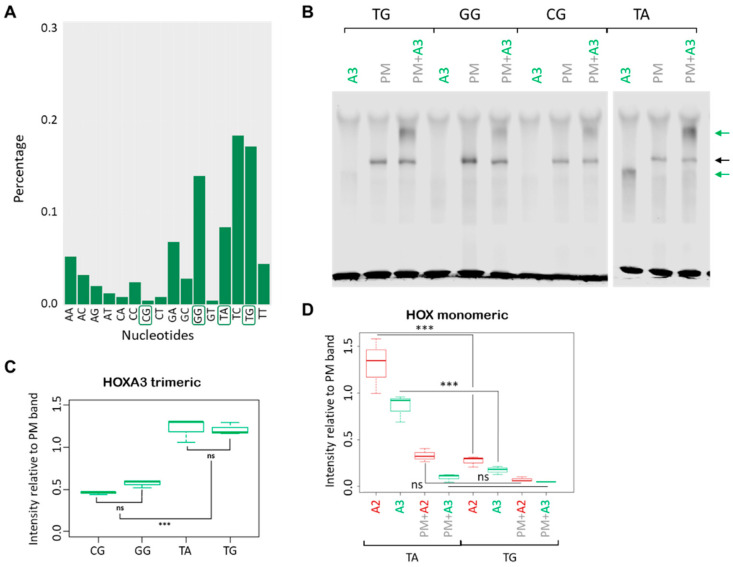
HOXA3 binding to *Meis2* probe variants. (**A**) Percentage of TGATNNAT variants (all possible permutations of the NN variable region) in top 250 HOXA3 peaks, ranked by fold enrichment. (**B**) HOXA3 binding to the *Meis2* probes containing mutations detailed in Figure 1B, in the presence and absence of PBX/MEIS (PM). Trimeric complex binding is reduced in GG and CG mutants, but TA mutant sees an increase in the binding of monomeric HOXA3. Black arrows indicate PBX/MEIS dimer, green arrows indicate HOXA3 interactions with the probe. (**C**) The quantification of HOXA3/PBX/MEIS bands from experiment in (**B**) (three replicates) shows a reduction in CG and GG mutants relative to the wild type, whereas TA is unchanged. All bands are quantified relative to the wild-type PBX/MEIS band. *** *p*-value < 0.0005; ns: no significant. (**D**) The quantification of HOXA2 and HOXA3 bands from the experiments in Figure 1C and Figure 2B (three replicates) shows an increase in the binding of monomeric HOX to the TA probe relative to the wild type.

**Figure 3 jdb-09-00055-f003:**
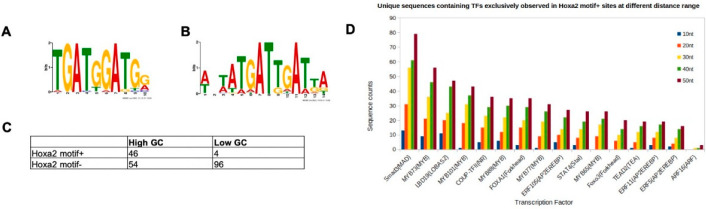
Properties of HOXA2 peaks containing TGATGGAT. (**A**,**B**) HOXA2 summits were divided in two sets on the basis of the presence of the TGATGGAT motif (motif+) or not (motif−). Regions were sorted by decreasing GC contents (within ±10 bp of the centre of the TGATNNAT motif) and searched for motifs in the top and last 100 GC-sorted sequences using MEME. (**A**) Regions with a higher GC content are enriched in TGATGGAT, (**B**) while regions with a low GC contain a HOX-PBX with a TG variable core. (**C**) The frequencies of TGATGGAT in the top 100 GC-high and last (low GC) 100 sites in the sorted list. (**D**) Motif occurrences in the flanking regions of motif+ versus motif− HOXA2 peaks identified by HOMER. Ratio of the number of unique sequences, for each TF, in both datasets (Hoxa2 motif+ and Hoxa2 motif−) using different distance ranges.

**Figure 4 jdb-09-00055-f004:**
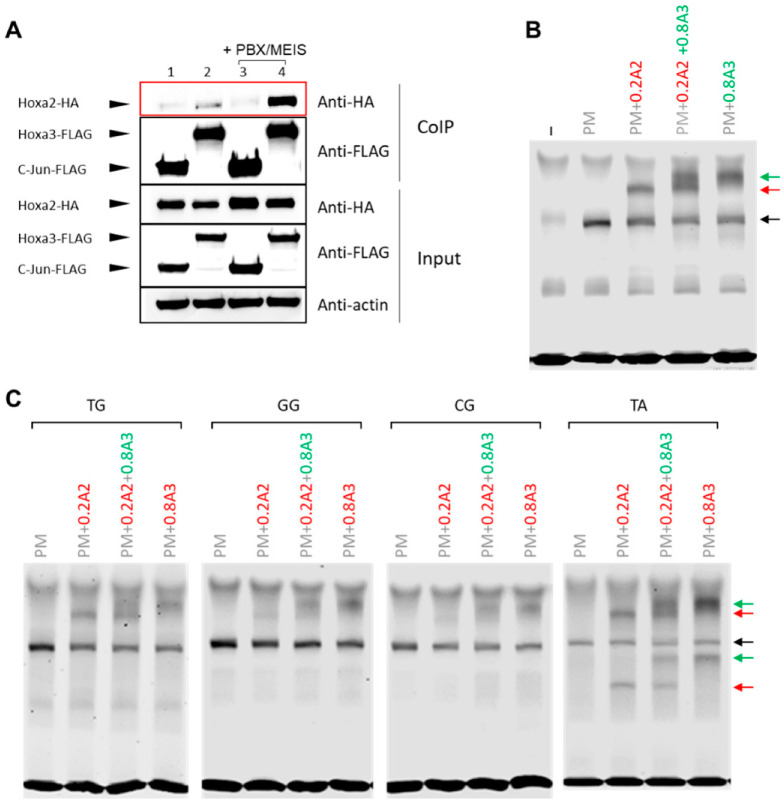
HOXA2 and HOXA3 form a complex with PBX and MEIS. (**A**) Co-precipitation assay. HEK293T cells were transfected with expression vectors for FLAG-and HA-tagged proteins. Proteins were analysed by western blotting before (input) and after (CoIP) co-precipitation using an anti-FLAG antibody. All conditions contain Hoxa2-HA in the presence of Jun-FLAG (1, 3), Hoxa3-FLAG (2,4); Meis1 and Pbx1 were added to conditions 3, 4. Jun represents a negative control. (**B**) HOXA2 and HOXA3 binding to *Meis2* probe when there is a 4x excess of HOXA3 and both are incubated for 5 min prior to the addition of PBX/MEIS (PM) and the probe. HOXA3 is unable to displace HOXA2. No tetrameric complex is observed. Black arrows indicate PBX/MEIS dimer, red and green arrows indicate HOXA2 and HOXA3 interactions with the probe, respectively. (**C**) Experiment in (**B**) repeated with the *Meis2* mutant variants detailed in Figure 1B.

## Data Availability

Not applicable.

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
