# Peer review of "Molecular Characterization of HOXA2 and HOXA3 Binding Properties"

_jdb, 2021, doi:10.3390/jdb9040055_

Round 1

Reviewer 1 Report

This paper by Bobola’s group describes differences in HoxA2 and HoxA3 binding differences to specific DNA sequences in vitro vs. in vivo, with or without the PBX-MEIS complex. The information provided are interesting albeit not very original and still very preliminary. Nevertheless they point out the importance of being very careful in assigning binding motifs to specific TFs, something totally missing in the various data banks. Moreover, they provide a possible mechanism explaining the differential ability of HoxA2 and HoxA3 in activating different districts of the embryo, despite switching off either of the two TFs synthesis.

I have some problems with the interpretations in the manuscript, because the DNA binding sites were identified by ChIPseq amalyses in vivo far less stringency for HoxA2 than for PBX-MEIS. In addition, the observation of an extensive overlap between HoxA2 and PPBX-MEIS binding sites was already reported in 2013, but the data are not even mentioned.

Author Response

We would like to thank the Reviewers for their helpful suggestions; we have addressed most of the issues that were raised by the Reviewers.

Reviewer 1

As requested, we have added a reference for the original paper describing the discovery of PBX and MEIS sites in HOXA2 ChIP-seq (Donaldson et al, 2012, Page 4). In this paper evidence of HOXA2 co-binding with PBX and MEIS was based on motif enrichment. In the following papers (Amin et al 2015; Bridoux et al 2020) we analysed the actual occupancy of PBX and MEIS in the same embryonic domain as HOXA2, which showed most of HOXA2 binds with PBX/MEIS.

Reviewer 2 Report

Mallen et al. “Molecular characterization of HOXA2 snd HOXA3 binding properties”

The manuscript by Mallen et al. deal with the important question how different Hox proteins can drive different transcriptional programs. To this end, the authors had previously compared in vivo binding of the tow Hox proteins HOXA2 and HOXA3, and now tested if sequence motif enrichment observed in vivo is explained by binding affinities in vitro. The authors found that there is a fundamental discrepancy between in vivo and in vitro binding and that the two Hox proteins heterodimerize, which might be an important mechanism to ensure posterior prevalence.

I recommend publication of this principally very interesting story after some points have been clarified and addressed.

Major points:

In the first part of the paper, the authors investigated whether differential sequence enrichment in in vivo identified peaks reflects divers binding affinity. To this end, they studied the binding abilities and affinities of HoxA2 and HoxA3 to the Meis2 probe, which contains a high confidence HoxA2 in vivo binding peak, using EMSAs. This analysis revealed that the core sequence strongest bound by HoxA2 in vivo (TGATGGAT) was not at all bound by HoxA2 alone in vitro, that binding could only mildly be increased by the addition of the two cofactors Meis and Pbx (formation of trimeric complex), although Meis and Pbx bound equally well as dimers to this and other sequences (with the exception of TGATTAAT). Intriguingly, this latter sequence, TGATTAAT, bound HoxA2 alone best in vitro, although this was not the case in vivo. I have to admit that all EMSAs shown are very small (Fig. 1, 2, 4), thus it is really hard to evaluate the shifts properly. This needs to be changed. In addition, I would suggest that for all forms (monomeric, dimeric and trimeric) a quantification of band intensity is performed. From the picture, it is unclear to me whether the addition of HoxA2 and P+M to the TGATCGAT version leads to a shift or not. If it does, it is minor, a quantification would clarify this. I also find the statement on page 4 that these results suggest “the intriguing possibility that HOXA2 binding on its own or in a complex with PBX/MEIS could mediate opposing outcomes …” too strong, this should be toned down. Another important point is the absence of the free probe without the addition of any protein in Figure 2 and 4. This is important to add, as in Figure 4B the free probe shows additional bands that also run on the same height when protein is added. This raises of course some general concerns about the bandshifts. Are these unspecific bands that are just increased by the addition of protein? Thus, these controls need to be added in Figure 1, 2 and 4C. The authors could also include some competitor experiments to show which bands are due to unspecific or specific protein binding.

The authors move from HoxA2 to HoxA3 (page 4, last paragraph), claiming that HoxA3 is more promiscuous based on their previous publication. Which sequences were enriched in HoxA3 in vivo peaks? How different are they to the HoxA2 peaks? It would be good if they mentioned this. The authors then test HoxA3 binding to the Meis2 probe and its variants. Here, the bandshifts “suffer” from the same lack of control and competition experiments.

To investigate if Hox2A and Hox3A are within one complex the authors use CoIP experiments displayed in Figure 4A. I have some difficulties to understand the differences between the first two panels and the rest. Could it be that there is some labelling missing, like “input”, “ IP”? Please adjust this otherwise this figure panel is not clear.

In general, the labelling and the figure legends could be improved, to explain for example what “PM” stands for, what the black, green and red arrows mean.

Minor points:

The term “moiety” in the first part of the introduction might be exchanged for terms like “domain, region, …” 

Author Response

We would like to thank the Reviewers for their helpful suggestions; we have addressed most of the issues that were raised by the Reviewers.

As requested, we have provided enlarged images of all EMSA experiments (Figs 1, 2 and 4). Additionally, we have provided the complete quantification results from these experiments. Quantification from Fig. 4B can be found in Fig. S2, and Fig. 4C quantification can be found in Fig. S3.

 We have toned down the statement “…suggesting the intriguing possibility that HOXA2 binding on its own or in a complex with PBX/MEIS could mediate opposing outcomes” to “It will be interesting to test if HOXA2 binding on its own or in a complex with PBX/MEIS could mediate opposing outcomes in gene expression (e.g. repression versus activation)”.

We thank the Reviewer for requesting appropriate controls for the bandshift experiments. Rather than including these negative controls in each image, we have provided a representative image (Fig S1A; incubation with PBX/MEIS is also shown as reference) demonstrating the binding pattern of wild type and mutant probes in the absence of proteins, and mentioned these results in the main text (Page 5-6). The results conclusively show that, with all the versions of the probe tested, addition of HOX, MEIS and PBX result in the formation of specific bands.

Competitions experiments were unfeasible in the time frame for the resubmission (10 days).

As requested the same analysis performed on HOXA2 has been performed on HOXA3 peaks to shows HOXA3 preferred sequences (Fig 2A).

We would like to thank you for bringing to our attention the missing labels in Figure 4A, which appear to have been lost during editing and are now restored.

Figure labels and legends have been updated to improve their interpretation and to be more consistent through the paper.

As suggested, the term “moiety” in the first part of the introduction has been replaced with “domain”.

Round 2

Reviewer 1 Report

no other comments.

Author Response

Thank you very much for participating in the peer review process of this manuscript

Reviewer 2 Report

All issues have been addressed, the manuscript is ready for publication.

Author Response

(The authors gave the same response as above.)
